# Method for Quantifying Variation in the Resistance of Electronic Cigarette Coils

**DOI:** 10.3390/ijerph17217779

**Published:** 2020-10-24

**Authors:** Qutaiba M. Saleh, Edward C. Hensel, Risa J. Robinson

**Affiliations:** 1Department of Computer Engineering, Rochester Institute of Technology, Rochester, NY 14623, USA; qms7252@rit.edu; 2Department of Mechanical Engineering, Rochester Institute of Technology, Rochester, NY 14623, USA; rjreme@rit.edu

**Keywords:** coil resistance, e-cigarette, electronic nicotine delivery system, pod style, atomizer

## Abstract

In electronic nicotine delivery systems (ENDS), coil resistance is an important factor in the generation of heat energy used to change e-liquid into vapor. An accurate and unbiased method for testing coil resistance is vital for understanding its effect on emissions and reporting results that are comparable across different types and brands of ENDS and measured in different laboratories. This study proposes a robust, accurate and unbiased method for measuring coil resistance. An apparatus is used which mimics the geometric configuration and assembly of ENDS reservoirs, coils and power control units. The method is demonstrated on two commonly used ENDS devices—the ALTO by Vuse and JUUL. Analysis shows that the proposed method is stable and reliable. The two-wire configuration introduced a positive measurement bias of 0.086 (Ω), which is a significant error for sub-ohm coil designs. The four-wire configuration is far less prone to bias error and is recommended for universal adoption. We observed a significant difference in the coil resistance of 0.593 (Ω) (*p* < 0.001) between the two products tested. The mean resistance and standard deviation of the reservoir/coil assemblies was shown to be 1.031 (0.067) (Ω) for ALTO and 1.624 (0.033) (Ω) for JUUL. The variation in coil resistance between products and within products can have significant impacts on aerosol emissions.

## 1. Introduction

### 1.1. Theoretical Foundation

Typical pod-style electronic nicotine delivery systems (ENDS) consist of three subsystems: the reservoir, power control unit (PCU) and lithium battery. The reservoir includes a mouthpiece, e-liquid storage compartment, a heating element herein called the coil, sometimes a wick, and an aerosol generation chamber, sometimes referred to as the atomizer. Reservoirs designed for re-use, permitting users to refill the reservoir with e-liquids, are sometimes called “open systems.” Reservoirs designed to be disposable, not intended to permit e-liquid refills by the user, are sometimes called “closed systems.” Most pod-style ENDS reservoirs are designed with the heating element, or “coil”, fully integrated with a wick to deliver e-liquid from the reservoir to the heating element such that thermally generated aerosol mixes with inhaled air for delivery through the mouthpiece to the user. Other ENDS reservoir designs may permit the user to replace the coil, wick, or adjust the flow path through the reservoir. The PCU contains electronics which manage the user interface, controls energy delivered to the pod, and withdraw or replace (recharges) energy to the battery. The lithium battery is the energy storage unit and provides energy to the PCU and the pod. The three subsystems interact with one another in an integrated manner. Variation in one component, such as the coil in the reservoir, may have different impacts on aerosol generation depending upon its interaction with the PCU and the battery.

The aerosol generation performance of an ENDS is jointly dependent upon the physical characteristics of the ENDS pod, PCU and battery [1,2,3,4,5,6], characteristics of e-liquid in the reservoir [2,3,5,7], and user behaviors of puff flow rate and puff duration [1,8,9,10,11,12]. Understanding the theory of ENDS operation elucidates potential effects of variation in resistance arising from interactions with the PCU and battery. All ENDS are fundamentally heat and mass transfer devices, which are commonly studied in engineering disciplines [13] for medical, industrial, residential and commercial products. The purpose of the heating coil is to convert electrical power discharge from the battery into thermal power dissipated inside the coil. The resulting thermal power (W) is distributed via heat conduction to the surface area of the coil as a heat flux (W/m^2^), which is then transferred to the surrounding air/e-liquid via heat convection. As the e-liquid solvent temperature reaches its saturation temperature (the effective boiling point of the e-liquid mixture), mass transfers from the e-liquid reservoir into the air stream to form an aerosol. The combination of surface heat flux, coil surface temperature, heat and mass transfer coefficients and e-liquid composition jointly affect the rate of aerosol generation. The aerosol generated at the interface between the coil and e-liquid experiences further changes as it progresses through the flow channel of the ENDS toward the user. While the effectiveness of aerosol generation is impacted by many factors, it is dominated by the electrical energy from the battery dissipated as thermal energy in the coil. The amount of heat energy, E, in Joules (J), dissipated in the coil is defined as the product of the instantaneous direct current power, P, supplied to the coil, in watts (1 W ≡ 1 J/Sec), and time duration over which the power is supplied (Sec). The power is a function of the electrical current, I_coil_ (A), flowing through the coil, the voltage, V_coil_ (V), applied across the terminals of the coil, and resistance, R_coil_ (Ω) of the coil itself as shown in Equation (1).
(1)$$\begin{array}{c} \begin{array}{c} P=V_{\mathrm{coil}}\;I_{\mathrm{coil}}=\frac{V^{2}{}_{\mathrm{coil}}}{R_{\mathrm{coil}}}\;=\;I^{2}{}_{\mathrm{coil}}\;R_{\mathrm{coil}} \end{array} \end{array}$$

The coil resistance is an inherent physical characteristic of the coil which depends primarily upon the composition and purity of the coil and its geometry. In this study, we include the internal electrical connections between the coil and the pod housing to quantify the effective resistance of the coil assembly. The voltage, V_coil_, and current, I_coil_, passing through the coil are related to one another using the classical definition of Ohm’s law, V_coil_ = I_coil_ × R_coil_, from physics [14]. The PCU controls the duration over which power is supplied to the coil. The PCUs employed in early ENDS designs simply shorted the voltage available from the battery across the coil for an interval of time. As the battery discharged over time, its available voltage decreased and hence the power delivered to the coil decreased. All modern ENDS PCU designs control the time duration of power delivery, while some PCU designs actively control the current, I_coil_, flowing through the coil and other designs actively control the voltage, V_coil_, applied across the terminals of the coil. Fully understanding the effects of variation in coil resistance, R_coil_, on aerosol emissions cannot be accomplished without understanding the logic implemented in the ENDS PCU.

Equation (1) shows that the desired power of an ENDS can be achieved by a specific ratio of coil resistance and applied voltage. In the lowest-cost ENDS designs, the applied voltage is limited by physical constraints of the most common lithium batteries, which peaks at approximately 3.7 (V) and decreases as the battery discharges. Higher-cost ENDS designs may actively control the output voltage using a “boost” converter [15], at the penalty cost of reduced operating time between recharging. ENDS manufacturers, over time, have sought to increase the power dissipation in the coil in order to increase the rate of e-liquid aerosolization. Given the physical limitations of low-cost rechargeable lithium batteries, ENDS designers choose to reduce the coil resistance as the most appealing parameter to increase power. This is illustrated by an example. Consider an ENDS designer who specified a power level of 12 (W) and a lithium battery operating at nominally 3.7 (V). Equation (1) dictates that a coil resistance of 1.14 (Ω) should be used. If power of more than 12 (W) is desired, even smaller resistance is required. Otherwise, a stack of two or more batteries connected in series or a DC-to-DC boost converter can be used to step up the applied voltage [15]. Using two batteries is not a desirable solution, because it increases the cost, weight and volume of the ENDS. Boost converters are increasingly common in modern ENDS but appear to be primarily used as a means to extend operating lifetime and maintain steady power while the battery discharges.

The desire for high-power ENDS pushes designers to increasingly use coils with low resistance values, which led to the introduction of the sub-ohm devices which use coils that have a resistance of < 1 (Ω) [16,17]. Reducing the coil resistance to 0.068 (Ω) permits instantaneous power as high as 200 W. The sub-ohm devices are reported to satisfy several features desired by users such as intense flavor, warm vape and big clouds, which is associated with high airflow that is suitable for the direct-to-lung inhale style [18,19]. Sub-ohm coils are mostly available in the box-mod ENDS style; however, some pod-style ENDS started to use coils with resistances of less than 1 (Ω) such as SMOK2 pod (0.8 (Ω) and NORD2 (0.3, 0.4, 0.6 (Ω)) [20], TARGET PM80 (0.2, 0.3, 0.6, 0.8 (Ω) [21], and Z-BIIP (0.48 (Ω)) [22].

Several studies investigated the effects of power values on the performance of ENDS devices while others focused on coil resistance values. As demonstrated with Equation (1), power levels can be controlled in real time by manipulating the applied voltage and current, or by installing a coil with a different resistance. The power level of an ENDS may be increased by increasing the applied voltage (which has the effect of increasing the current) or by decreasing coil resistance values while keeping the applied voltage constant. The same approach can be followed for decreasing power levels. For this reason, most of the results achieved from the studies focusing on power and voltage values can be generalized to coil resistance values with appropriate adjustments and vice versa. However, it is essential that studies investigating the effect of power on emissions document both the power dissipation and coil resistance in order to make results generalizable to other products.

### 1.2. Context of Prior Work

Previous studies have shown that coil resistance affects both the amount of vapor generated and constituents. Cirillo et al. 2019 [23] showed that reducing coil resistance leads to higher concentrations of some carbonyls and reactive oxygen species (ROS). Their research also reported that vapor generated by coils with lower resistances has a higher negative impact on cell viability. The same research group [24] also showed in a separate work in 2019 that the production of selected aldehydes increased as coil resistance decreased from 1.5 (Ω) to 0.25 (Ω). The effects of the aerosols generated by the two coils on Sprague–Dawley rats was studied. The rat group exposed to the 0.25 (Ω) vape showed disorganization of alveolar and bronchial epithelium. The same group also showed higher perturbation of the antioxidant and phase II enzymes compared to the 1.5 (Ω) groups. Gillman et al. 2016 [25] studied the effects of changing power on the total yield mass and the formation of aldehyde. Their results showed that power has significant impact on the concentration of aldehyde in the vapor. Although their main focus was power, they used several coils with various resistance values to control the power. This indicates that in this study, the physically important factor is resistance values, as already demonstrated in Equation (1). Chausse et al., in 2015 [26], suggested coil resistance could be the key to lung toxicity. Their analysis showed that the combination of certain voltage and coil resistance values may lead to a high impact on human health. Hiler et al. 2019 [27] investigated the effects of changing heating coil resistance on nicotine delivery, puffing topography, subjective effects and liquid consumption. They used off-the-shelf coils with two resistance values of 0.5 (Ω) and 1.5 (Ω) which are supposed to consume a power of 40.5 (W) and 13.5 (W), respectively. Hiler reported that lower-resistance coils were found to deliver higher nicotine and have higher liquid consumption. Several prior studies did not document the method used to measure coil resistance or assess variation in this key parameter. Cirillo et al. 2019 [23,24], Sleiman et al. 2016 [28], Ogunwale et al. 2017 [29] and Soulet et al. 2018 [30] reported coil resistance as a study parameter without documenting the method used to measure resistance. Researchers may be tempted to select off-the-shelf coils with resistance values that suit their study. This approach assumes that the coil resistances reported by the manufacturer are accurate, neglects manufacturing variation between coils, and may limit study designs to ENDS brands that offer coils with different resistance values.

Conversely, other studies reported the measurement methods employed. Gillman et al. 2016 [25] used a measurement instrument that is specialized in milliohm resistance measurement (Extech milliohm meter, 380560) which claims high precision and low error rates. Hiler et al. [27] used an off-the-shelf ohm meter “Coil Master 521 TAB v2”, advertised to test homemade coil resistance and ensure proper operation before yse. The Coil Master [31] is reported to have a 510 threaded connector, compatible with many box-mod ENDS, and is equipped with a fire button which applies voltage to the coil during test. The manufacturer reports that readings have an error of approximately +/− 0.05 (Ω), which corresponds to a reading error of +/− 10% when the measured coil has a resistance of 0.5 (Ω). The high error margin of this meter suggests that it is generally not appropriate for use in scientific studies.

In summary, many studies have been performed to assess the impact of power and coil resistance on emissions and toxicity, but no studies have been performed to evaluate the accuracy of the methods used to measure coil resistance. Many modern commercially available box-mod ENDs are equipped with a resistance meter to report coil resistance to the user, but no studies could be found that validate the reliability of these values.

With the current trend to reduce coil resistance values to sub-ohm levels [16,17], manufacturing variations might have greater effects on the actual resistance of the coils, which in turn may give rise to variation in the composition of emissions from the ENDS. To date, we have found no reports that quantify such variation nor how it might affect the expected performance of the ENDS. Previous inconsistencies in reporting coil resistance may be caused by the absence of a robust standard method.

### 1.3. Study Objectives

This study focuses on demonstrating accurate and robust methods for quantifying: (1) the effective resistance of the heating coil in a fully assembled pod, (2) variation in effective coil resistance across pods from a single manufacturer, (3) the repeatability of the test method and apparatus, and (4) differences in effective coil resistance between manufacturing designs. The proposed methods are demonstrated on two popular [32,33,34] commercially available pod-style ENDS: ALTO by Vuse [35] and JUUL [36]. The test fixtures described herein are supported by detailed published protocols [37], enabling researchers to use commercially available test equipment and instructions to fabricate custom test fixtures for measuring the effective coil resistance of integrated pods from various manufacturers.

## 2. Materials and Methods

### 2.1. Constant Current Resistance Measurement Method

The constant current method is commonly used for resistance measurement in conjunction with a digital multimeter (DMM). Modern DMMs for laboratory use incorporate an integrated voltmeter and constant current source. The DMM constant current source, I_Source_, is used to supply current to the device under test (DUT) while measuring the voltage, V_Voltmeter_, across the DUT. The resistance of the DUT, R_Measured_, is determined as the ratio of measured voltage over the applied current, consistent with Ohm’s law:(2)$$R_{Measured}\;=\;\frac{V_{Voltmeter}\;}{I_{Source}}$$

Two configurations are commonly used with the constant current method. The most common two-wire configuration is appropriate for general purposes with large DUT resistances on the order of kilo-ohms and mega-ohms. The four-wire configuration is more appropriate for accurate measurement of sub-ohm coil resistances. The effect of measurement configuration is assessed herein by using both configurations on a sample of pods from two manufacturers and conducting a repeated-measures difference between configurations.

#### 2.1.1. Two-Wire Configuration

The schematic shown in Figure 1 is known as the two-wire configuration [38]. Two terminals (denoted as the + and − terminals) are supplied with a constant current, I_Source_, from the DMM. two-wire leads are connected from the terminals of the DMM to the opposing ends of the DUT coil being tested. Kirchoff’s voltage law states that the voltage drop across three components in series is the summation of the voltage drop across each component in the series. Thus, the voltage, V_Voltmeter_, measured by the DMM is the summation of the voltage drops across the DUT and the two-wire leads. Kirchoff’s current law asserts that the steady current flowing through three resistive components in series are equal. Consequently, the measured current through the DUT will be identical to the current through the wire leads. Equation (3) shows the calculations in detail, which can be simplified to the second form if we assume that the two-wire leads are of the same composition, diameter and length.
(3)$$\begin{array}{c} R_{Measured}=\frac{V_{Voltmeter}}{I_{Source}}=\frac{V_{Lead}+V_{Dut}+V_{Lead}}{I_{Source}}=R_{DUT}+2\times R_{Lead} \end{array}$$

The resistance of wire leads can vary widely between research laboratories and test benches, easily between 0.010 (Ω) and 1.000 (Ω) depending mainly on their materials and lengths. The resistance of the wire leads, R_Lead_, introduces a significant bias error between the DMM observed, R_Measured_, and the actual coil, R_DUT_, when the DUT has low resistance. For example, when measuring the resistance of an ENDS coil with a true resistance of 1 (Ω) in this configuration with two-wire leads that have a resistance of 0.050 (Ω) each, the measured value would be approximately 1.1 (Ω), a 10% bias error in measurement. If we use the same apparatus and wire leads to measure sub-ohm coils, the percent of bias error would be correspondingly higher. The wire leads are the main source of error in this configuration. The bias error can be reduced by using very short wire leads with high electrical conductivity (e.g., gold instead of copper). However, short leads are often difficult to manipulate in the lab, and this approach does not completely remove the bias introduced by the wire leads. The four-wire configuration offers a practical and robust approach to removing the bias error associated with the two-write configuration.

#### 2.1.2. Four-Wire Configuration

The four-wire configuration employs two current wire leads (force + and force −) to supply current through the DUT and two separate sensor wire leads (sense + and sense −) to measure the voltage across the DUT. The four-wire configuration eliminates the bias effect of lead resistances described in the two-wire resistance measurement configuration [38]. Figure 2 shows the schematic of this configuration. The voltmeter has very high resistance (on the order of megaohms) and thus very low current (on order of picoohms) flows through the sense +/− wire leads. Thus, the voltage drop across the sense +/− wire leads is negligible and the voltage measured by the voltmeter is the same as the voltage cross the DUT. The resistance of the wire lead is totally insignificant, and the measured resistance is unbiased compared to the two-wire configuration. Random errors such as those associated with analog-to-digital conversion in the DMM remain in both configurations but are negligible in comparison to the quantities typically required for characterization of ENDS coils.

### 2.2. ENDS Product-Specific Test Fixture

An apparatus was assembled to conduct constant current method resistance measurements in both the two-wire and four-wire configuration. The DMM used for all resistance measurements in this study was a Model 34465A from KEYSIGHT™ [39] and supports both the two-wire and four-wire resistance configurations with several user-selectable ranges (100 (Ω) to 1000 (MΩ)). The DMM was set to the “auto scale” option for all observations conducted in this study, resulting in every observation being measured on the 100 (Ω) range. The DMM was connected to a desktop computer running the Microsoft Windows™ 10 operating system (Microsoft, Redmond, WA, USA) with a USB-2 cable. A custom MATLAB™ script (MathWorks, Inc., Naatick, MA, USA) was used to trigger measurements and collect readings from the DMM and save the data to comma separated value (CSV) test files for later analysis.

While commercial off-the-shelf (COTS) four-wire leads, commonly called “Kelvin leads”, may be used to connect the DUT to the DMM, we elected to fabricate custom four-wire leads which were permanently soldered to each DUT fixture. We determined that COTS Kelvin leads might not be the best option for measurement of ENDS coil resistance. In some types of ENDS, the coils are separable from the reservoir and wick. However, during and after usage, the coil is in contact or submerged with the e-liquid. In other types of ENDS, such as the pod-style device studied here, the coil is permanently attached within the pod and is submerged in the e-liquid reservoir. In removable pod ENDS devices, the pods most often make electrical contact with the ENDS PCU using two different style connectors. Several pod-style ENDS employ two spring-loaded connectors, also known as pogo pin connectors, for the + and − terminals on the PCU side while mating with two corresponding flat connectors on the pod side. The connectors come in various sizes and shapes across ENDS designs and, most importantly, they have different distances between the +/− terminals. Additionally, the mechanical means of retaining the pod in the ENDS is different from one manufacturer’s design to another. Some manufacturers use a friction fit, some use a magnetic clasp, and others may use detents. This variation between ENDS designs means that the pod-style coil terminals are not directly exposed and cannot directly connect to COTS Kelvin lead clips. Further, we desired a test fixture which would allow us to assess the impact of the pod-retaining mechanisms employed in various ENDS designs on the repeatability of effective coil resistance measurement. Electrical contact resistance caused by the connectors is a potential confounder in natural environment operations of ENDS products, and is worth investigation.

Accordingly, this study incorporates a special holding fixture which is unique to each ENDS product and retains the pod with its integrated coil in position while measurements are made with the DMM. Each holding fixture uses a scavenged ENDS PCU housing and original manufacturer’s connectors to mimic the housing, connectors and pod retention employed in the original ENDS to create a more realistic setup and produce accurate resistance measurements. This study employs connectors and PCU housing of the same design as the ENDS device under test. Figure 3 shows the holding fixture built for an ALTO-style ENDS with a snap pod retainer as part of its internal structure. The Model 34465A KEYSIGHT™ DMM [39] is used in conjunction with each unique ENDS holding fixture. Each test fixture is secured vertically using a table-top vise with the pod housing on the top end while the four wires are to the bottom. This setup made it easy to position the test fixture and switch between pods during the experiment.

The process has currently been demonstrated on two ENDS designs. Detailed tutorials for building resistance measurement holding fixtures are available at [37]. The process of building the pod holding fixture includes multiple steps which are briefly summarized here:Discharge the battery of the ENDS prior to opening the device.Open the ENDS PCU to access its internal structure. Use standard electrical safety precautions when working in the presence of possible charge carrying components such as capacitors. Avoid shorting any electrical leads during disassembly.Remove the battery and the readily accessible PCU electronic components.Locate the internal side of the connectors. These spring connectors are used to connect the PCU to the pod or tank section of the ENDS. The spring connectors may be directly soldered to a printed circuit board (PCB) as in the JUUL, or indirectly connected to the PCB via thin wires as in the ALTO.Solder four lead wires, force +/− and sense +/−, to the ENDS PCU +/− connectors, respectively. Care must be taken not to damage the connectors or gaskets which lie between the ENDS PCU PCB and the END PCU pod receiver. Details of these connections are important in developing an accurate and robust fixture and are discussed in detail and with photographic guides in [37]. In the case of the device built for ALTO, the four lead wires are soldered to the cut end of the manufacturer’s thin wires linking the connectors to the PCB, taking care to protect the connector and surrounding plastics case from soldering heat. These two thin wires will be added to the measured coil resistance in addition to the resistance of the connectors themselves. These extra resistances can be measured and subtracted from coil resistance or can be simply neglected if they appeared to be very small. This point is discussed in detail in the Section 3. These thin wires are inherently present in the ENDS circuitry to supply power to the coil. Their resistance is added to the coil resistance contribute to the total resistance seen by the ENDS PCU. Some designs of ENDS PCU may rely on these thin wires to dynamically sense coil resistance. Thus, including the thin wires in the testing apparatus circuitry represents an accurate measure of the effective resistance of the pod/coil assembly.Make a small groove at the end of the ENDS PCU housing to make room to pass the four wires out from the fixture to the DMM.Connect color-coded banana plugs to the free end of the four lead wires for inserting into the DMM.

### 2.3. Data Sampling Procedure

Each pod to be tested is marked with a unique identification number. Prior to making a resistance measurement, this ID number is entered into the data logging script. The operator places the pod to be tested in the holding fixture and presses a button on the computer to initiate data sampling. The script has the ability to read the resistance using the two-wire and four-wire modes without additional operator intervention. The tested pod remains inserted between the two-wire and four-wire measurements. The script can read and report a single observation of coil resistance or can read and record 120 sequential readings of the same pod taken at one-second intervals, to evaluate the performance of the fixture and validate stability. Each data reading is tagged with the pod ID, two-wire vs. four-wire configuration, trial number (for test/re-test repeatability), a time-date stamp (to identify 120 sequential readings), and the numerical value of the resistance reported by the DMM. All readings are made in the same lab under the same environments including the same storage condition and room temperature.

### 2.4. Test Specimens

Two popular pod-style ENDS devices were chosen for this study. The Vuse ALTO [35] and JUUL [36] ENDS are two of the most popularly used e-cigarettes especially among teenagers [32,33,34]. We purchased *N* = 22 Vuse ALTO pods and *N* = 16 JUUL pods filled with nicotine flavor e-liquid with a manufacturer-reported 5% nicotine concentration. The manufacturers report that ALTO pods are filled with ~1.8 mL of e-liquid [35], while JUUL pods are filled with ~0.7 mL of e-liquid [36]. All pods were new and in the manufacturer’s original sealed packaging until opened for this test. Pods were purchased from local retail brick-and-mortar establishments and national online vendors. We observed that the Vuse ALTO pods were identified with *N* = 5 unique manufacturing lots, and the JUUL pods were identified with *N* = 2 manufacturing lots.

### 2.5. Statistical Analyses

Descriptive statistics were computed for each ENDS design including the mean, median, interquartile range and outlier analysis. Standard statistical tests were used to assess all data collected. Assessment of bias evident between the two-wire and four-wire configuration of the constant current resistance test method was conducted with a repeated-measure (single-sample) t-test for each ENDS design studied. Assessment of the one-to-one intra-class correlation coefficient, ICC_1:1_ [40], was conducted in the same lab using the same testing apparatus to assess the repeatability of the results when the same sample of ALTO and JUUL pods were individually tested, and then re-tested after a 5 months interval of time. A total of 120 independent readings using the four-wire configuration of each uniquely identified pod were taken on two different days, separated by five months between. The test/re-test correlation coefficient, r_test/re-test_ [40], was conducted to test the consistency or reliability of the measurement. Ten readings using the four-wire configuration were taken for each pod and the pod was removed from the test fixture and inserted back between consecutive readings with a 2–5 s interval. The one-to-one intra-class correlation coefficient, ICC_1:1_ [40], and test/re-test correlation coefficient, r_test/re-test_ [40], were computed and reported for each ENDS studied. Assessment of differences in mean effective coil resistance between ENDS designs was conducted using a two-sample t-test under the assumptions of a normal distribution, unequal sample sizes and unequal variances. The assumption of normally distributed samples was evaluated using a quantile–quantile (Q–Q) plot of data observations vs. theoretical normal distribution. Assessment of manufacturing variation was conducted using a normal distribution with point estimates for the mean and standard deviation to predict the ±6σ range of coil resistances anticipated for mass-produced ENDS pods.

## 3. Results

Coil resistance test fixtures are built for two pod-style products: ALTO and JUUL. The constant current resistance measurement method was used to measure *N* = 22 ALTO pods and *N* = 16 JUUL pods with both the two- and four-wire lead configurations. A sampling distribution of the mean was conducted for each test specimen, consisting of 120 repeated-measure resistance readings with the digital multimeter at one-second intervals. Variation between repeated readings exhibited a 95% confidence interval of less than 0.0002 (Ω) for every set of 120 repeated observations per pod and test configuration (2 wire vs. 4 wire), indicating excellent stability of the test fixture.

The mean resistance value of the *N* = 22 ALTO pods had a range of 1.018–1.304 (Ω) for the two-wire configuration and 0.933–1.214 (Ω) for the four-wire configuration. The mean resistance of the *N* = 16 JUUL pods had a range of 1.631–1.744 (Ω) for the two-wire configuration and 1.544–1.659 (Ω) for the four-wire configuration.

Figure 4 shows a box plot of four groups of data: the two-wire and four-wire readings for ALTO pods and JUUL pods. The mean resistance and standard deviation of *N* = 22 ALTO pods was observed to be 1.118 (0.053) (Ω) using the two-wire lead configuration and 1.031 (0.052) (Ω) using the four-wire lead configuration. The ALTO data exhibits a slight positive skew and a paired t-test between the two-wire lead and four-wire lead configurations exhibits a difference in means of δ = 0.087 (Ω) (*p* < 0.001). The mean resistance (and standard deviation) of *N* = 16 JUUL pods was observed to be 1.710 (0.032) (Ω) using the two-wire lead configuration and 1.624 (0.033) (Ω) using the four-wire lead configuration. The JUUL data exhibits a negative skew and the paired t-test between the two- and four-wire configurations exhibits a difference in means of δ = 0.086 (Ω) (*p* < 0.001). Results demonstrate that the two-wire lead configuration exhibits a positive bias of ≈ 0.087 (Ω), as would be expected due to the additional resistance (2 × R_Lead_) present in the two-wire lead test configuration illustrated in Figure 1. For this reason, only four-wire configuration data is taken to the next analysis step.

The test/re-test correlation coefficient, r_test/re-test_, and the one-to-one intra-class correlation coefficient, ICC_1:1_, were computed to assess the repeatability of the testing apparatus. The one-to-one intra-class correlation coefficient is conducted on two sets of readings. The first set is the means of the 120 readings reported for in the previous paragraph. The second set consists of repeated measurements for the same pods five months later. This comparison will show the stability of the test fixture and pods over a period of five moths. The ICC_1:1_ for ALTO (*N* = 13) is 0.9997 (13) *p* < 0.001 and for JUUL (*N* = 16) is 0.9960 (16) *p* < 0.001. The test/re-test correlation coefficient is used to show the consistency or reliability among measurements. A set of 10 consecutive readings with a 2–5 s interval is taken for each pod. The test/re-test correlation coefficient for ALTO (*N* = 17) is 0.9997 (144) *p* < 0.001 and for JUUL (*N* = 16) is 0.9873 (135) *p* < 0.001.

A quantile–quantile (Q–Q) plot of each data sample obtained using the unbiased four-wire lead configuration is shown in Figure 5. The data was evaluated against several standard distributions, and the normal distribution was found to be the closest fit to both ALTO and JUUL data, with a slight positive and negative skew, respectively, consistent with the box plots in Figure 4.

Figure 6 shows a normalized histogram and fitted pdf for the four-wire resistance readings of the ALTO and JUUL. The plot shows that both pod products exhibit manufacturing variations in coil resistance. The mean and standard deviation of the sample distributions from the unbiased four-wire lead configuration are 1.031 (0.067) (Ω) for the ALTO and 1.624 (0.033) (Ω) for the JUUL. A two-sample t-test was conducted to assess the effective mean resistance between the two pod products, demonstrating a result of δ ≈ 0.593 (Ω) (*p* < 0.001).

## 4. Discussion

The four-wire constant current method is the preferred technique for quantifying electronic cigarette coil resistance. Between-product comparisons may not be reliable when resistance measurements are taken using the popular two-wire method. Thus, the popular two-wire method should be avoided in future studies. For example, in our controlled setting, the two-wire method introduced a bias of 0.086 (Ω), which was 15% of the observed difference between product means of 0.593 (Ω). If care is not taken with the wire lead resistance, the bias can be even larger than 15% and obfuscate important variation between products. Similarly, if two-wire comparisons are made between data collected by two different labs with two different apparatus, we may inadvertently conclude that there is no significant difference between product characteristics or ascribe a difference to the product which could actually be a result of the test apparatus. While two-wire resistance measurement methods are common in many laboratory settings, they may not be sufficiently accurate for measuring the resistance of electronic cigarette coils with a resistance of ≈3 (Ω) or lower.

Results demonstrated that a sample of coils from a single manufacturer procured across manufacturing lots exhibit variation, which is a significant fraction of the nominal coil resistance. Such manufacturing variation is expected. The amount of manufacturing variation in coil resistance associated with a particular product may have significant implications on the emissions resulting from the use of the product. The coefficient of variation (standard deviation over the mean) was observed to be 5.1% for ALTO and 2.0% for JUUL. Using the sampling distribution of the mean, it is reasonable to infer that the ±3σ coil resistance of ALTO and JUUL pods vary by as much as ±15% and ±6% for ALTO and JUUL, respectively, when considering the true population of large production lots typical of a national and global distribution channel.

Differences in coil resistance, whether associated with bias error from the two-wire configuration, variation between products or variation within a single product, may yield important variations in the emissions of total particulate matter (TPM) and presence of hazardous and potentially hazardous constituents (HPHCs) in the emissions. Furthermore, the impact of variability in coil resistance in the pod (or e-cigarette reservoir) on emissions is closely related to the electronics used in the power control unit (PCU) of the e-cigarette. The PCUs of modern e-cigarettes are far more sophisticated than a simple battery. However, examining the effect of variation in coil resistance on power dissipated in the coil by a simple direct current (DC) circuit of a battery across a coil provides insight into the joint impacts of both the coil and the PCU on emissions.

Consider an electronic cigarette with a nominal coil resistance of 1 (Ω) (measured using four-wire configuration) and powered by a battery with a fully charged voltage of 3.7 (V). To illustrate the point, consider the PCU as being a simple “on/off” switch with no active voltage or current control and no supporting pulse width modulation. The nominal power dissipated will be $P=\frac{3.7^{2}}{1.0}\;=\;13.69\;\left(W\right)$, as given by Equation (1) and the nominal current flowing through the coil will be *I_coil_ =V_coil_ / R_coil_ = 3.7/1.0 = 3.7* (A), as determined by Ohm’s Law, Equation (2). It is well known from DC circuit analysis that the total energy delivered from the coil to the liquid is limited by the product of the nominal power and duration of activation. Similarly, it is well known from heat and mass transfer analysis that the rate of mass transfer from the liquid to the aerosol stream is affected by the surface area, flow path, and flow rate. As the power, *P*, and energy increase, we can anticipate more emission of TPM. As the current, *I_coil_*, increases, the temperature of the coil itself will increase through a well-known phenomenon known as “Ohmic heating.” We might anticipate, then, that increases in current flowing through the coil might give rise to increased production of HPHCs in the emissions resulting from thermal decomposition of e-liquid constituents. Thus, understanding the influence of variation in the coil resistance of e-cigarettes is essential to understanding emissions.

We demonstrated that the two-wire configuration introduced a positive bias, overestimating the true value of the coil resistance by 0.086 (Ω). If we use this biased estimate of coil resistance, we would underestimate the power, $P=\frac{3.7^{2}}{1.086}\;=\;12.6\;\left(W\right)$, and current, *I_coil_ = 3.6* (A), in the coil. These biased estimates are not conservative, and could very well give rise to apparent inconsistencies observed between the emissions produced when comparing products to one another.

The same concern holds true when we assess the effect of manufacturing variation between coils of the same product design. A manufacturing variation of +/− 15% in coil resistance with a simple PCU would result in variations in coil power and current dissipation of −13%/+18%. As the manufacturing variation in coils increases, the potential adverse consequences of changes in HPHC emissions also increases. It has been reported that coils with lower resistance values may have higher negative health impacts [23,24,25,27]. Therefore, it is essential to develop a full understanding of the manufacturing variation in coil resistance associated with electronic cigarettes. This high difference in the expected instantaneous power has the potential to drastically change the performance of the device, constituents of the aerosol produced, and the e-liquid consumption rate.

Electronic cigarette power control units (PCUs) which employ active voltage control may mitigate the adverse impact of variation in coil resistance. Both the ALTO and JUUL are equipped with PCUs exhibiting active control logic. The sophistication of PCUs varies widely between product designs. The potential of the PCU to mitigate manufacturing variation in coil resistance deserves further attention. Little is known about the performance of various PCUs and their limitations.

The robust method introduced herein provides a foundation for investigation of several research questions, including the following. What resistance variation is exhibited between coils of different products? What variation in resistance is exhibited within a product design? What are the effects of resistance variation on the aerosol composition and generation rate? How does the resistance of a coil change over the course of its operating life? How does the resistance of a coil change during a vaping session? How do these variations affect the performance, lifetime and safety of the lithium battery? What is the relation between coil resistance, power, and the performance of the PCU? To what extent can the PCU overcome variations in coil resistance? What is the effect of using interchangeable coils (such as the now common 510 threaded reservoir) in conjunction with PCUs on emissions? How does this framework inform potential adverse health consequences of product misuse and product hacking?

## 5. Conclusions

The preferred constant current resistance measurement method using four-wire leads is demonstrated to provide stable, accurate, repeatable and unbiased observations of the resistance of pod-style electronic cigarette coil assemblies. The commonly employed two-wire lead configuration is demonstrated to introduce a positive bias, the magnitude of which is dependent upon the laboratory testing apparatus and impedes reproducing results between independent laboratories. The constant current four-wire lead method is recommended as the standard method for measuring the resistance of electronic cigarette coil assemblies. The coil resistance measurement method, which was demonstrated using two brands of pod-style electronic cigarettes, is broadly applicable to other brands and styles of electronic cigarettes.

A quantile–quantile assessment demonstrated that the manufacturing variation in the effective coil resistance of pod assemblies for two brands of popular electronic cigarettes is normally distributed. The mean coil resistance of pods varies significantly between the brands of electronic cigarettes tested, demonstrating that the sample mean resistance of the ALTO pods was 0.593 (Ω) lower than the mean resistance of JUUL pods (*p* < 0.001). Furthermore, the sample of ALTO (*N* = 22) pods tested exhibited a mean and 99% confidence interval of 1.031 ± 0.0405 (Ω), while that of the JUUL (*N* = 16) pods exhibited a mean and 99% confidence interval of 1.624 ± 0.0243 (Ω). The resistance measurement method is thus valuable for assessing variations between brands of electronic cigarettes and for quantifying manufacturing variation which may be anticipated within a single product brand. Products exhibiting larger manufacturing variation in coil resistance may result in the wider variability of emissions generated from those products. As a result, it is recommended that the distribution of manufacturing coil resistance (using the constant current four-wire lead method) be reported in future comprehensive emissions studies and new product applications for coils and coil assemblies.

## Figures and Tables

**Figure 1 ijerph-17-07779-f001:**
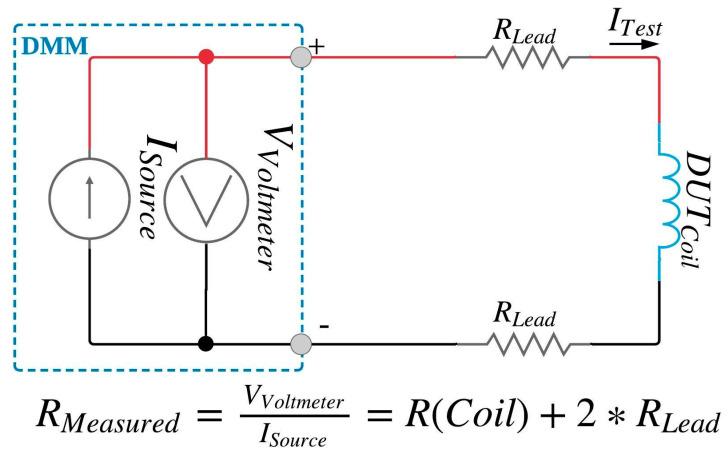
Two-wire resistance measurement schematic. Dotted box labeled digital multimeter (DMM) is a simplified version of internal schematic of the digital multimeter.

**Figure 2 ijerph-17-07779-f002:**
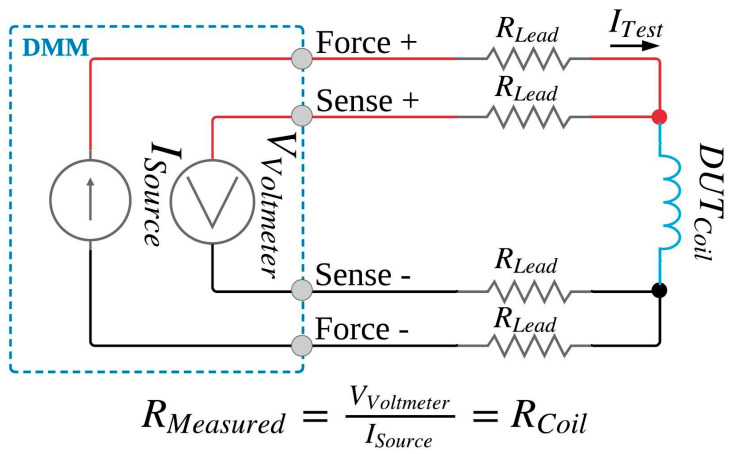
Four-wire resistance measurement schematic. Dotted box labeled DMM is a simplified version of internal schematic of the digital multimeter.

**Figure 3 ijerph-17-07779-f003:**
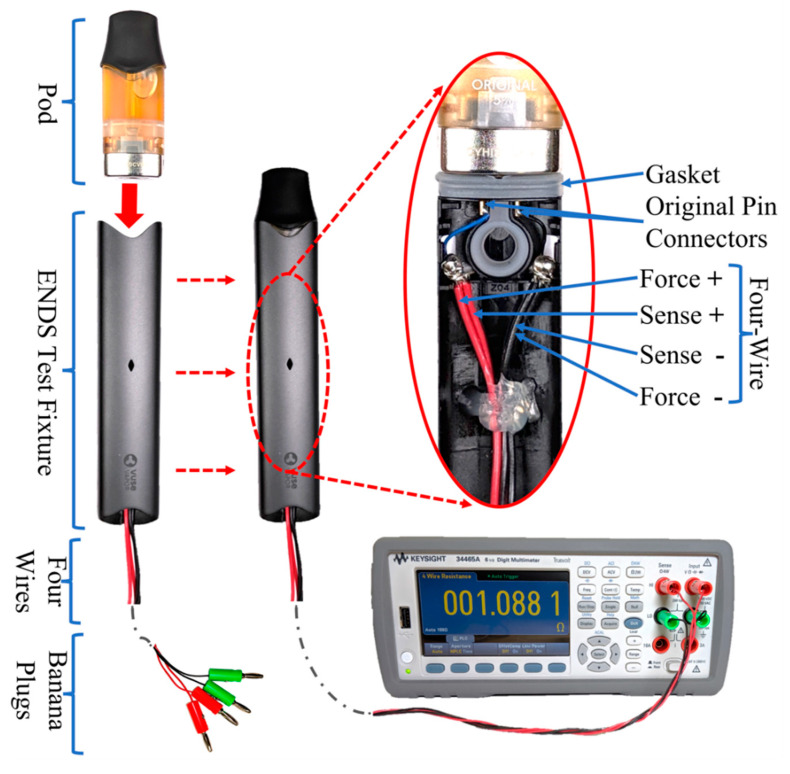
Custom electronic nicotine delivery systems (ENDS) test fixture for Vuse/ALTO pod-style ENDS coil resistance measurement in conjunction with a Keysight Model 34465A digital multimeter. The exploded view shows an inside view of the wire connections.

**Figure 4 ijerph-17-07779-f004:**
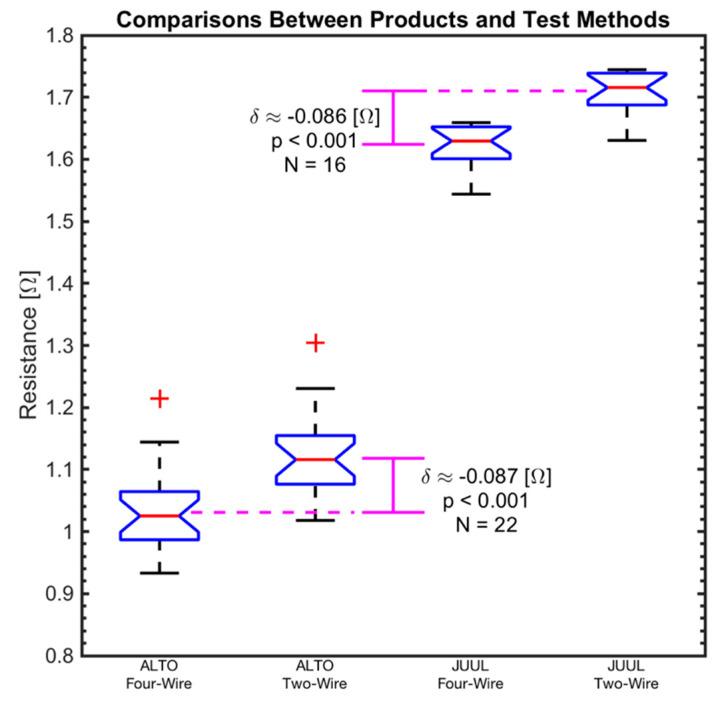
Box plot and t-test results of ALTO (*N* = 22) and JUUL (*N* = 16) coil resistance readings for the two-wire and four-wire configurations using the constant current method. The differences between the two-wire and four-wire configurations are 0.087 (Ω) (*p* < 0.001) and 0.086 (Ω) (*p* < 0.001) for ALTO and JUUL, respectively. The magenta difference bars are pointing at the group means while the red line in the box plot refers to the group medians. The box notches illustrate the 95% confidence interval on the median.

**Figure 5 ijerph-17-07779-f005:**
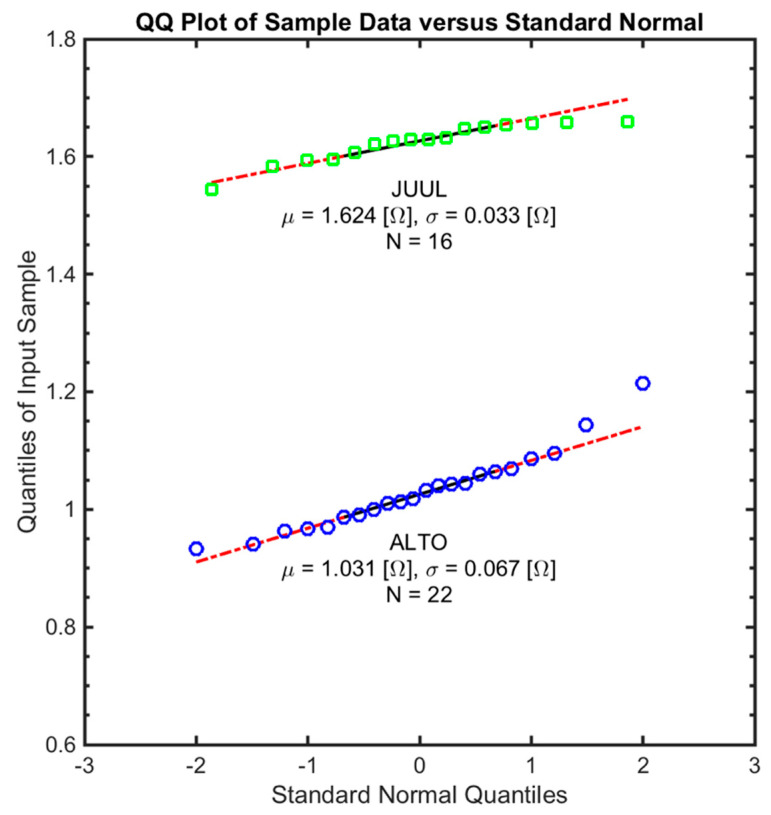
Q–Q plot of the four-wire coil resistance readings of ALTO (*N* = 22) and JUUL (*N* = 16) for a standard normal distribution. The data was tested against several standard random distributions and normal distribution was found to be the closest fit.

**Figure 6 ijerph-17-07779-f006:**
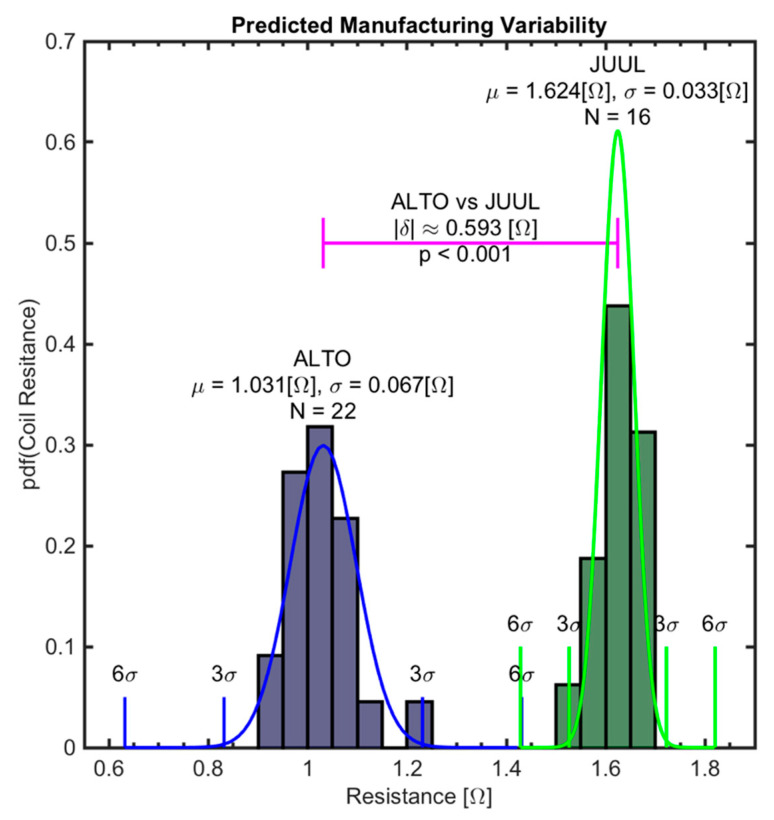
Normalized histogram and fitted Probability Density Function (PDF) for four-wire coil resistances of ALTO (*N* = 22) and JUUL (*N* = 16). The group mean resistance of ALTO pods is 0.593 (Ω) (*p* < 0.001) less than that of JUUL pods. The sample of ALTO pods tested exhibits more manufacturing variability than the sample of JUUL pods.

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
