# Peer review of "Method for Quantifying Variation in the Resistance of Electronic Cigarette Coils"

_ijerph, 2020, doi:10.3390/ijerph17217779_

Round 1

Reviewer 1 Report

It would be better if more products were tested and included.

Author Response

Thank you for your thoughtful and positive review of our manuscript.

We do indeed intend to use the method introduced herein to assess multiple Electronic cigarettes. We have another manuscript in development which applies this method to the assessment of 13 unique electronic cigarette designs.

Reviewer 2 Report

Method for quantifying variation in resistance of electronic cigarette coils by QM Saleh et col

General

This article is a technical article exploring the measurement of the resistance of the coils of two types of industrial cigarettes which are in the bosom of the tobacco industry the Alto de Vuse / Reynols and the Jull / PMI
This article does not touch on the subject of e-cigarettes with independent clearomizer on which the whole introduction of the article focuses to show the importance of the subject.
This article only deals with the applied power in a very accessory way, because in real life it is the association of power applied to the resistance, surface of the resistance, resistance that determines the temperature of the coil and therefore the danger of forming aldehydes and other toxic substances. The possible interest of the article is therefore industrial and is only very incidental to doctors or decision-makers. It's a bit like studying in depth the rotational speed of a motor vehicle and not the speed of the automobile.
The repeatability of the measurement on the same batch and the repeatability with devices from different batches (although the data must allow it to be done) is not reported in the article. The repetition of the measurement on the same device is described, on the same batch is not described, the repetition of the measurement 5 months later has a poorly described methodology). If the measurement is perfectly reproducible on the same device, but changes when the batch is changed, then this is an industrial problem in the manufacture of the devices. All this data is essential in a technical article of this type or else the title should be changed to say that it is a pilot for developing a method, but not to present the results as final results. More so as there was no interlaboratory measurement to qualify the measurement method.

Detail

Abstract

The authors claim that the study is robust, accurate and unbiased but the elements they provide are weak, in particular no repeatability study according to batches and reproducibility between laboratories
In particular, it is said in the abstract that the positive bias of the measurement of 0.086 ohms is significant for subohmic coils (widely used in some countries).

It is not clear in the summary to know what the standard deviation of the resistance measurement is (series of 120 measurements at one second of device of different batches) which is 3 times larger for the ALTO 1.031 + / - 0.067 (6.5%) than for JULL1.624 +/- 0.33 (2.0%)

Introduction

L31
Many electronic cigarettes have a clearomizer and not a pod, however presented as the only solution for e-cigarettes in the introduction.
L51
The detailed description of the vaporization lacks the condensation of the vaporized gas in fine droplets which gives the visible appearance of the emissions
L63
The most important is the composition, the purity comes only in a more incidental way
L61
One element missing from the equation if the goal is to prevent the formation of aldehyde is the surface area of the resistor in contact with the liquid. Indeed, the same power delivered by the resistance to the liquid is multiplied by 3 if the surface of the resistance in contact with the liquid triples.

109
The chapter on the context takes up imprecise studies which can only be taken into account when we know the applied power, resistance area and resistance in ohms, which is generally not done in studies. It is indeed necessary to apply an excessive power or look at the surface and resistance of the coil. It is noted that in these articles we rather accuse the power, whereas it is necessary to have the 3 parameters surface, resistance and power to define the conditions of occurrence
L128
Globally, intrinsically accusing resistance of undesirable effects is not based on anything, as your reference 26 repeats, it is the resistance (ohm and surface) and power pair that determines the risk of overheating (production of toxic aldehydes) or lack of heating (No aerosol production. Saying "Lower resistance coil were found to deliver higher nicotine and have higher liquid consumption" not to mention power misleads the reader
L159
Ok for the objective of measuring reproducibility for the same type of device from a manufacturer (2) and repeatability (Measurement and device) with the same type of resistance

L182 L214 L261 L348 L379 L382 L387
error message in the middle of the line (a priori sending to a figure)

e the same measurements
L320

There are only 2 lots for the Jull which does not allow us to study the difference between lots
L 339

Why did you choose to measure more Alto than Jull?
L 345
The variations in measurement, for example, of the Alto range from 1.18 to 1.304 ohms which is a large variation. Is there a defense between the lots, is there a difference with the temperature of the room is it is not known if it has been controlled
L369
Reproduction is done in the same lab by repeating the measurements 5 months later (to be described in the method chapter)

L375
It is not clear how the test was done again. The initial measurements turned off made on 120 measurements with one measurement per second, here we are talking about 10 consecutive reading with 2 to 5 seconds intervals. Please describe and explain precisely in the method chapter.
L 437
ok here you introduce the notion of surface which would also have its place in the introduction.
L417
An inter-laboratory study would be necessary to confirm this
L454
Low resistances are associated with high power and we can as well say that it is the high powers which expose to more formation of aldehyde, if we want to be in conformity with reality we must each time remember that this is the torque power resistance (with area) which must be known to calculate the risk
L463
Ok the list of unanswered questions remains wide, but in real life acrolein with the very unpleasant taste is a good element to avoid inhaling aldehyde which in real life is otherwise less than the quantity or exposes the cigarette of tobacco. Only under unrealistic experimental conditions can more be

Bibliography

No remark

Figures

Are useful

Author Response

Thank you for reviewing our manuscript and providing your useful notes and suggestions. Your feedback is valuable to us and has been considered while revising our manuscript. Responses to each of your points are provided in the attached word document, along with an indication of how the manuscript has been revised. We hope that our edits address each of the items raised and have improved the manuscript.

We have used the word "track changes" feature in our revised manuscript to make the edits easily identifiable.

We Thank you for your kind and thoughtful input.  Your review has given helped up improve the manuscript and we very much appreciate your perspective.
